# Blood L-Lactate Concentration as an Indicator of Outcome in Roe Deer (*Capreolus capreolus*) Admitted to a Wildlife Rescue Center

**DOI:** 10.3390/ani10061066

**Published:** 2020-06-20

**Authors:** Elena Di Lorenzo, Riccardo Rossi, Fabiana Ferrari, Valeria Martini, Stefano Comazzi

**Affiliations:** 1Department of Veterinary Medicine, University of Milan, 26900 Lodi, Italy; elena.dilorenzo@studenti.unimi.it (E.D.L.); valeria.martini@unimi.it (V.M.); 2Piacenza Wildlife Rescue Center, 29120 Niviano di Rivergano, Italy; info@piacenzawildlife.org (R.R.); fabi1981@virgilio.it (F.F.)

**Keywords:** roe deer, blood parameters, prognostic factors, blood lactate concentration, biomarkers

## Abstract

**Simple Summary:**

Roe deer are among the most frequent wild animals admitted to rescue centers in Italy. Reasons for admission include trauma, predation, starvation, and imprinting, with the final aim of hospitalization being full recovery and release into the wild for all cases. An accurate triage procedure is vital for predicting the outcome, to avoid unnecessary time spent in captivity, and an excessive allocation of time and resources on animals with a minimal chance of recovery. Since lactacidosis is often associated with death in hospitalized animals, and has been associated with poor prognosis in humans and domestic animals, we proposed an evaluation of blood L-lactate using a rapid whole blood test in order to predict the outcome of hospitalized roe deer. A cut-off of 10.2 mmol/L was identified as the best, in order distinguish animals with minimal chance of surviving and release. For these animals, humane euthanasia should be considered as an option.

**Abstract:**

Roe deer (*Capreolus capreolus*) are among the most frequent patients of rescue centers in Italy. Three outcomes are possible: natural death, euthanasia, or treatment and release. The aim of the present study is to propose blood L-lactate concentration as a possible prognostic biomarker that may assist veterinarians in the decision-making process. Sixty-three roe deer, admitted to one rescue center in the period between July 2018 and July 2019, were sampled and divided into 4 groups according to their outcome: (1) spontaneous death (17 cases), (2) humanely euthanized (13 cases), (3) fully recovered and released (13 cases), and (4) euthanized being unsuitable for release (20 cases). In addition, blood samples from 14 hunted roe deer were analyzed as controls. Whole blood lactate concentrations were measured with a point of care lactate meter. Differences among groups were close to statistical significance (*p* = 0.51). A cut-off value of 10.2 mmol/L was identified: all the animals with higher values died or were humanely euthanized. The results suggest that roe deer with lactatemia higher than 10.2 mmol/L at admission, have a reduced prognosis for survival during the rehabilitation period, regardless of the reason for hospitalization and the injuries reported. Therefore, humane euthanasia should be considered for these animals.

## 1. Introduction

The roe deer (*Capreolus capreolus*) is an artiodactyl mammal belonging to the Odocoileinae subfamily—the only Euro-Asian member—and is the smallest European cervid. Roe deer are widely distributed in Europe, with the exception of northern Scandinavia and some of the islands, notably Iceland, Ireland, and the Mediterranean Sea islands. In Italy, an estimated population of more than 426,000 roe deer has been reported [1]. Roes are among the most common cervid species in Italy [2], thus they are often recovered by wildlife rescue centers. The reasons for their hospitalization are quite variable and principally include traumas (car accident or lawnmower trauma), predation, starvation, and imprinted animals, classed as being dangerous to humans or fenced into urban areas. There are three possible outcomes for hospitalized animals in a wildlife rescue center: (1) natural death, (2) euthanasia, and (3) treatment, rehabilitation, and release into the wild. An accurate and rapid triage is crucial for the veterinary staff of the centers in order to avoid prolonged treatment of animals with minimal chances of release. 

Biomarkers are specific tests used to monitor normal or disease processes [3]. Such laboratory tests may help veterinary clinicians with challenging decisions regarding initial triage and management in an objective fashion. One medical complication in rescued wild animals is metabolic acidosis due to lactacidemia related to trauma, shock, capture stress, and myopathy [4]. Thus, we hypothesize that the concentration of L-lactate in blood, from roe deer admitted to a wildlife rescue center, may be used to predict outcome, irrespective of the cause of injury. The aim of the present study was to test our hypothesis and select a suitable cutoff value for L-lactate concentration in the blood of injured roe deer, to identify animals that will not survive rehabilitation and release. 

## 2. Materials and Methods

For the purposes of the present research, blood samples from roe deer of different sexes and ages referred to the Piacenza Wildlife Rescue Center, from July 2018 to July 2019, were collected. The reasons for admission varied. 

All animals were sampled for diagnostic purposes in order to better frame the clinical case (except for blood samples from the control group that were taken from hunted dead animals immediately after shooting). According to the guidelines of the authors’ institution, formal approval from the Ethical Committee was not required (EC decision 29 October 2012, renewed with the protocol n° 02–2016).

The individuals included in the study are divided into classes depending on sex and age, which were determined based on morphological aspects. As far as age is concerned, three different classes can be identified: fawns, yearlings, and adults. Yearlings can be recognized by their light build and slender body. Young males have antlers which may have up to two tines, whereas females have a slender profile, as their abdomen is not relaxed by gestation. A more precise evaluation of age was obtained following a dental examination to determine the state of eruption and wearing down of the teeth.

All admitted roe deer were divided into 4 groups according to their final outcome: (Group 1) spontaneous death during the recovery period, (Group 2) euthanized for welfare reasons (i.e., animals near death where euthanasia was considered as the only possible option to reduce pain and suffering and), (Group 3) fully recovered and released roe deer, and (Group 4) animals euthanized for other reasons (i.e., animals that were judged unable to survive rehabilitation and release back into the wild). In addition, as a control (Group 5), blood samples from 14 hunted roe deer, from the same area, were taken from the cardiac cavity immediately after shooting death.

In general, a manual restraint during the recovery period (animals blindfolded and hogtied) was preferred to sedation, as chemical restraint would invalidate appropriate triage of polytraumatized animals. In the rare cases when sedation was performed, two different protocols were used: dexmedetomidine (10 µg/kg, Dexdomitor, Zoetis, Rome, Italy) + ketamine (2 mg/kg, Ketavet100, MSD) + methadone (0.2 mg/kg, Semfortan, Dechra, Turin, Italy), if the roe deer require surgery, or dexmedetomidine (10 µg/kg) + ketamine (2 mg/kg) + butorphanol (0.2 mg/kg, Dolorex, Intervet, Peschiera Borromeo, MI, Italy), if only chemical restraints, diagnostic procedures and minor surgery were required. 

All samples were taken from the cephalic vein within 2 h of arrival to the Center, put into EDTA coated tubes, and L-lactate concentrations were evaluated within 10 min from sampling by means of a portable point of care lactate meter (Accutrend Plus, Roche, Monza, Italy). The instrument measures the concentration of L-lactate in whole blood using a reactive strip via a bench top clinical chemistry analyzer system, which has been validated in cattle as reliable and linear up to 16.6 mmol/L [5]. This method was preferred to classical plasma spectrophotometric evaluation using a clinical chemistry analyzer since it is portable, practical, rapid, and cheap and it does not require plasma separation, which is sometimes difficult in the absence of an equipped laboratory.

### Statistic Analysis

Descriptive statistics were calculated. Data from hunted deer were used to calculate reference ranges according to the official guidelines [6]. The Dixon method was applied to identify and eliminate the outliers [7].

A Shapiro–Wilk test was performed to assess normal distribution of data among outcome groups. A Kruskal–Wallis test was performed to assess possible differences in L-lactate concentration among outcome groups. Contingency tables were prepared, and the Pearson’s chi-squared test was performed to assess possible differences in sex, age, and cause of admission among outcome groups.

ROC curves were drawn, and coordinates were used to determine the cutoff value of L-lactate concentration, having 100% specificity and the highest sensitivity in the identification of animals with a negative outcome. These criteria were applied to select the cutoff value, since we aimed to identify animals with a negative outcome and avoid false positive results.

At first, survivors and animals who encountered spontaneous death were included in the ROC curve analysis. Thereafter, humanely euthanized animals (group 2) were also included and grouped together with those who died spontaneously.

All statistical analyses were performed by means of a dedicated software (SPSS 20.0 for Windows). Significance was set at *p* ≤ 0.05 for all analyses. 

## 3. Results

A total of 77 blood samples with L-lactate concentrations were available and divided into five different groups: (1) spontaneous death (17 cases), (2) humanely euthanized (13 cases), (3) fully recovered and released (13 cases), (4) euthanized for other reasons (20 cases), and (5) control animals (14 cases). A total of 30 females and 47 males, of which 38 adults, 16 subadults (10–23 months old) and 23 fawns (less than 10 months old), were tested. Among hospitalized roe deer, the prevalent causes of admission were trauma (45 cases), followed by predation (9 cases), imprinted animals (1 case) and other causes (8 cases). The four groups were homogeneous as for gender, age, and cause of admission (*p* > 0.05 for all analyses).

Results of blood lactate concentrations in different groups of roe deer are reported in Table 1 and Figure 1. Differences among groups were not significant (*p* = 0.051).

The reference range obtained after exclusion of outliers in our control group was 2.7–5.7 mmol/L.

Based on ROC curve coordinates, an L-lactate cutoff of 10.2 mmol/L was selected as the best to classify survivors and animals having encountered spontaneous death, with a 100% specificity and 47.1% sensitivity in detecting animals with a negative outcome.

When animals euthanized for welfare reasons (Group 2) were included in the analysis, an L-lactate cutoff of 10.0 mmol/L was selected as the best one, with a 100% specificity and a 46.7% sensitivity in detecting animals with a negative outcome. The aforementioned cutoff of 10.2 had a 100% specificity and a 43.3% sensitivity.

## 4. Discussion

Blood L-lactate derives from anaerobic glycolysis and in physiologic conditions, a production of 0.8 mmol/h/kg has been demonstrated, in humans, to contribute to obtain a normal plasma concentration of less than 1 mmol/L [8]. Concentration may increase with increased anaerobic metabolism due to many different causes, including tissue hypoxia, shock, increased gluconeogenesis, sepsis, anemia, or muscular overwork and damage. Blood lactate concentration has been proposed as a possible prognostic test in emergency conditions in both humans and animals [3,9]. A reduced prognosis in critical human patients has been reported with persistent high lactacidemia [9]. Furthermore, another study [10] suggested that lactacidemia higher than 4.5 mmol/L is correlated to death in 78% of critical patients with emergency conditions. Similarly, blood L-lactate concentration influences prognosis in horses with complicated colic [11], and in critically ill neonatal foals [12]. The use of a portable point of care lactate meter has also been validated in equine medicine [13], as a useful support to veterinary clinical decision making.

In wild cervids, plasma L-lactate has been evaluated as part of a laboratory panel to check the effects of capture using different methods [14,15], in different conditions [16], using different immobilizing drugs [17,18] or premedication [19]; however, to the best of our knowledge, no data about the possible role of plasma lactate concentration, to define the health status and predict the outcome of captured animals, have been published. Reference intervals for free ranging wild animals are often lacking. 

In the present study, the analysis of samples from 14 hunted roe deer (control Group 5) provided some reference to compare the concentration of plasma L-lactate from recovered animals. All roes in the study derived from the same area of study and were killed by selector hunters with a single shot when they were calm and minimally alert; blood samples were immediately taken via intracardiac puncture after being shot and were rapidly transported to the center for analysis. For these reasons, we consider this group as an adequate control for defining plasma L-lactate concentration in free ranging roe deer in the study area. However, the authors acknowledge that these control animals’ elevated L-lactate levels were likely associated with acute death, and that these values do not represent those of normal, healthy roe deer. In addition, cardiac blood could have a lower L-lactate concentration in comparison with venous peripheral blood, especially if oxygenated. In comparison, the reference range obtained in our control group was 2.7–5.7 mmol/L, lower than previously reported for 22 trapped roe deer (5.1–18.9 mmol/L) [11]. This supports the hypothesis that capture itself induces an increase in plasma L-lactate concentration. Therefore, for all the above-mentioned reasons, in the present research, we preferred comparing values from rescued roe deer with different outcomes, instead of using this reported reference interval. Interestingly, in this control group, a couple of samples showed higher L-lactate concentrations than the reference interval (8.4 and 15.3 mmol/L), and in one case, it was also higher than the prognostic cutoff values identified. This result supports the high variability of this parameter and confirms that plasma L-lactate concentration cannot be used as a stand-alone parameter to predict prognosis, especially when interpreting single admission L-lactate values, rather it should be considered in the context of clinical examination. In addition, the higher values in some of the control deer, could suggest that the control group was not 100% fit and healthy and/or had suffered from stress, dehydration, blood loss or other causes of elevated L-lactate, since we do not have any information about timeframes between the shooting and actual death of each animal. 

If we considered hospitalized roe deer, in which the modality of rescue, immobilization and handling are consistent, we found that the overall ranges of plasma L-lactate concentration are similar to those reported in previous studies for trapped animals [15,19], further supporting the possible effect of capture stress. 

The cutoff value of 10.2 mmol/L was chosen as a negative prognostic indicator, representing deer that are most likely destined to spontaneous death or require humane euthanasia. We selected a cutoff value with 100% specificity, where few, if any, animals with L-lactate above this level would survive, to avoid under-treatment or euthanasia of animals with a possible chance of surviving. This high cutoff value is responsible for the low sensitivity found. The limited sensitivity allowed us to identify less than half of the deer with a poor prognosis, but we cannot state that the low L-lactate concentrations are necessarily associated with a good outcome. This choice was made since, from a clinical point of view, we preferred to avoid any under-treatment (or euthanasia) in roe deer with a chance of survival. We are aware that this value may induce an overestimation of the chances of survival for some animals, but we consider it an acceptable risk from an ethical perspective. If we also apply this cut-off value to Group 4 (animals euthanized for other reasons) we found that 5 cases out of 20 (25%) showed L-lactate concentrations higher than the value of 10.2 mmol/L, thus supporting euthanasia in these animals. The median concentration of L-lactate found in non-surviving deer compares well (although it is a little lower) with those found at admission in one study of horses with colonic volvulus (non-survivor: 9.1 mmol/L) [20] and in another of dogs with gastric dilation and volvulus (non-survivor: 7.9 mmol/L) [21]. However, the cutoff value selected in the present study is higher than those used in the above-mentioned studies (6.0 mmol/L and 7.4 mmol/L, respectively) since, as already mentioned, we made the choice of using 100% specificity for predicting a bad outcome in roe deer. To our knowledge, no specific studies on the prognostic value of L-lactate in roe deer or other cervids are available to date.

The present study has some limitations. First, the limited number of cases did not allow us to stratify cases according to age and cause of admission. Second, plasma L-lactate concentration was evaluated at arrival at the rescue center, even though the time from being found to arrival at the center was variable according to different aspects, which may have biased our results. However, blood withdrawal was not always possible at the time of animal discovery or injury, (since it would require veterinary professional intervention in all the cases) and therefore, we chose to use the time of first admission to the center to best standardize and compare results. Third, few of the roe deer were sedated to allow for easier transportation and to decrease stress. However, this procedure was not standardized, since it depended upon the clinical condition, type of problem, and the presence of a veterinarian, and it was not possible to revisit which cases received sedation. Premedication with acepromazine has been reported to possibly decrease capture stress and myopathy, and reduce L-lactate concentration, thus increasing survival [19]. However, to the best of our knowledge, no data are available regarding the possible effects sedation could have on L-lactate concentration. Importantly, in terms of prognosis, this would not affect the value of L-lactate concentration for predicting outcome. 

In addition, in the present work we also added a control group from hunted roe deer who were shot dead as a comparison. The authors are aware that this group is probably not ideal for the creation of a reference interval, since it is biased by many different conditions and also influenced by different methods of blood withdrawal (cardiac in control deer vs peripheral blood in other deer). However, obtaining a reference interval in healthy wild animals is often challenging, since many blood analyte concentrations are strongly influenced by capture stress, and the use of a control group kept in captivity is probably not adequate as a comparison. Despite these limitations, the authors speculate that results from hunted deer may be useful for comparison since they were lower than the trapped roe deer reference value. 

Lastly, our work only included L-lactate values at admission and did not evaluate serial L-lactate in these animals, which has been reported to be more valuable than a single admission value when determining a prognosis [22,23].

Future studies should assess possible fluctuations of L-lactate concentrations over the time between admission and release and their possible prognostic role. In addition, it would be interesting to validate the same test and define accurate cutoff values for other types of samples, such as capillary blood taken from an ear-edge prick, which is minimally invasive and does not require a veterinary professional. Considering the limited volume of blood required for the test (50 μL), such a small amount of blood could be sufficient for such testing.

## 5. Conclusions

The data obtained in the present research support the use of plasma lactate concentration at admission to predict outcome of recovered roe deer, independently from the cause of recovery. Plasma lactate may be easily evaluated using a point of care portable analyzer in a very rapid, cheap, and feasible way. The results obtained may be interpreted by veterinary professionals in conjunction with other clinical information to predict prognosis, thus avoiding directing excessive efforts and resources on untreatable animals, and reducing over- and mis-treatments. Furthermore, our results will help guide euthanasia as a humane solution for wild animals facing little chance of release back into the wild.

## Figures and Tables

**Figure 1 animals-10-01066-f001:**
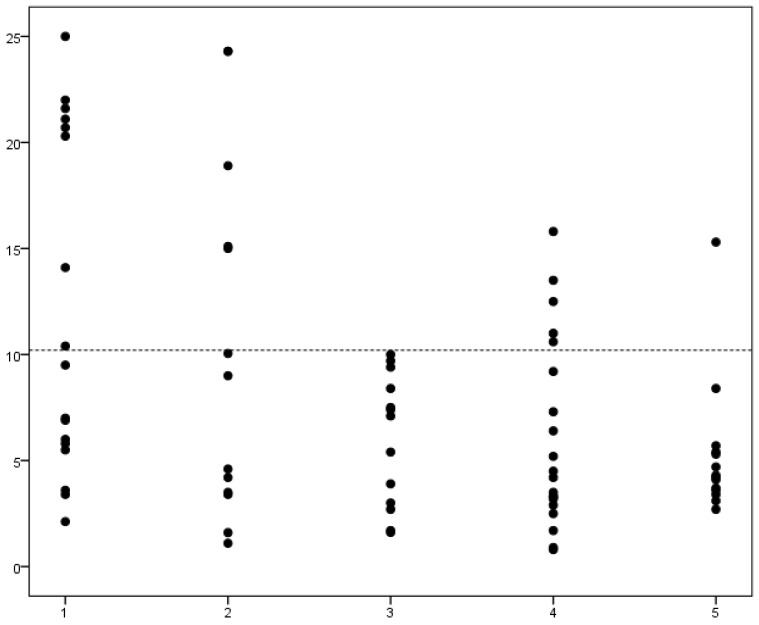
Dotplot showing results of blood L-lactate concentration in 77 roe deer. *X*-axis, outcome after hospitalization: (Group 1) spontaneously dead; (Group 2) euthanized for welfare reasons; (Group 3) recovered and released; (Group 4) euthanized for other reasons; (Group 5) hunted deer. *Y*-axis, L-lactate concentration (mmol/L). Dotted line: cutoff selected based on ROC curve coordinates, having a 100% specificity in discriminating between Group 1 and 3.

**Table 1 animals-10-01066-t001:** Results of descriptive statistics on blood L-lactate concentrations among different groups of roe deer. 1 No statistical differences were detected amongst groups (*p* = 0.51). # number of cases.

Group	Whole Blood L-lactate Concentration (mmol/L)		Cases with L-lactate > 10.2 mmol/L	
	Median	Min	Max	# (%)
1: spontaneous death (*n* = 17)	9.5	2.1	25.0	8/17 (47.1%)
2: euthanized for welfare reasons (*n* = 13)	9.0	1.1	24.3	5/13 (38.5%)
3: recovered and released (*n* = 13)	7.1	1.7	10.0	0/13 (0%)
4: euthanized for other reasons (*n* = 20)	4.3	0.8	15.8	5/20 (25%)
5: hunted deer (*n* = 14)	4.3	2.7	15.3	1/14 (7.1%)

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
