# Peer review of "Blood L-Lactate Concentration as an Indicator of Outcome in Roe Deer (Capreolus capreolus) Admitted to a Wildlife Rescue Center"

_animals, 2020, doi:10.3390/ani10061066_

Round 1

Reviewer 1 Report

Thank you for making all the changes from your original submission. I appreciate the hard work that must have taken, but this now read really well and it was worth the effort. I have two comments:

1) I must apologise for having missed this on the first reading, but I do not  think the figures in your 'Results' add up. Having counted and recounted your figures  I am unsure if there were 71 or 77  samples included, for these reasons.

On line 81 it says fawns were not included in the study but on line 136 they seem to be. If they were included change line 81.

Line 132 suggests 71 blood samples, but the numbers in the different groups add up to 77.

Line 135 suggests 30 females and 51 males which would clearly be 81 animals.

Line 135 also suggests 32 adults, 16 subadults and 23 fawns so 71 animals.

Line 137 suggests there were 63 hospitalised animals plus the 14 controls which would be 77 in total.

I am struggling to count how many dots there are in Figure 1, is it 71 or 77? Table 1 has 77 sample animals included. All the figures need to add up to the correct number of samples, either 71 or 77. If data is missing (e.g. an animal did not have its sex or age determined) then just say that, everyone will understand how hard these things are to always do in the field.

2) I am still unsure that all the references are correctly numbered and referenced. For example;

On line 174 reference 11 is included which is an equine reference but the text would suggest human cases are being discussed

On line 178 reference 14 is used - the reference is for deer, but horses are being discussed.

Line 182 includes a deer reference 20 but reference 20 is equine.

I have not looked further or checked them all. Please check ALL references really really carefully before resubmitting.

I do not need to see this again myself, but have requested that the journal editor checks these things as well before publication.

Author Response

We wish to thank the reviewer for the very accurate revision. Starting from his/her points we carefully re-checked the number of cases and found that some cases have been omitted during the preparation. Now the numbers has been corrected in the text and the tables. The number of dots in figure 1 may not correspond to the real number of cases since two or more cases may have the same L-lactate concentration and they are represented as a single one.

Also the reference list have been re-checked. We found that during revision process, some numbers were wrong. Now we accurately correct each single reference.

Thank you

Reviewer 2 Report

Dear Authors,

Thank you for the revisions of your manuscript... I believe it is much improved.

However, I still have some concerns regarding your interpretation of your data.  I have checked with our statistician.  First, it is acceptable to use an ROC comparison even when no significance was found among groups. However, we are concerned that you did not include all of the groups (1-4) in the ROC curve analysis for L-lactate cutoff determination.  If you only select to compare select groups you risk mis-representing the population and do not have enough numbers for such comparison.

I see that you have now compared groups 1(spontaneous death) and 2(euthanized near death) with group 3... Why did the authors not include group 4(euthanized other reasons) as well in this ROC comparison?  Please include group 4 in the ROC comparison or otherwise defend your choice.

I have also made several other changes to the manuscript (see attached tracked changes) to improve language and make it more concise.

I believe your manuscript will be a valuable contribution to the literature with these improvements.

Author Response

We wish to thank the reviewer for the suggestions. All the corrections in the text have been accepted.

We wish to thank the reviewer for the useful suggestions and corrections on the text. All the corrections have been accepted and changed

Unfortunately, we disagree with the suggestion of including animals of group 4 in those of group 2 for calculating ROC curves. ROC curves need the study population to be definitively divided into “positive” and “negative” cases. In our study, we first included deer encountering spontaneous death (“positive” cases) and survivors (“negative” cases) and identified the cutoff with 100% specificity and the highest sensitivity as possible. As he reviewer also stated, this approach affected the number of animals included in the analysis. This is why we thereafter included in the “positive” group also deer having been euthanized. These patients were euthanized because considered close to death, but actually there is no way to know whether would they have died spontaneously or not: that’s why we felt not confident in including them in the “positive” group at first. However, the second round of analyses (when more animals were included) confirmed the performances of the cutoff selected. We think including animals euthanized for other reasons would be misleading, as the choice to perform euthanasia was not linked only to their health status at admission, and this would create a bias when creating ROC curve. For instance in group 4 also animals with severe trauma to a leg were included. These deer were euthanized since they were considered not suitable with release into wild but an amputation of the leg could be compatible with recovery. In these cases the concentration of L-lactate is not correlated to the final outcome and the inclusion of such a case in the ROC evaluation would create a strong bias.   

Thank you for the very accurate revision again. We think that the manuscript greatly improved.

Reviewer 3 Report

The paper has been improved and now it is suitable for publication.

Author Response

The authors wish to thank the reviewer for the useful suggestions

This manuscript is a resubmission of an earlier submission. The following is a list of the peer review reports and author responses from that submission.

Round 1

Reviewer 1 Report

I will attached an annotated document with many ‘post it notes’ attached with comments. I have repeated also the main points below.

Several comments around the language used in some places. I apologise for this as the authors’ English is excellent generally and way better than my Italian!

Line 12: Avoiding unnecessary time in captivity for the animal is another reason for rapid triage

Line 20: should ‘release’ be ‘treatment and release’? Just ‘release’ alone might suggest for example letting a trapped deer go at the site of entrapment.

Lines 24 and 24: the categories are hard to understand – see comments (66-67) later and on annotated document

Line 41: a reference for Roe deer being the most frequently presented animal would be good, even if just pers. comm. or ‘unpublished data’

Line 45: Recover and release might be better as ‘3) treatment, rehabilitation and release to the wild’

Lines 66-67: Does ‘1) spontaneously dead’ mean presented dead or died shortly after admission or similar? Can that be  clear? Category 2 might be better as ‘immediately euthanased for welfare reasons’

Line 74: I think the site of sample collection needs to be included. This is for two reasons first intracardiac sampling might affect lactate levels compared to peripheral venous sampling (see discussion comments). Second if sampling could also be from a needle pick  (e.g. ear or foot) it might make the technique useful for non-veterinarians (suggest include this in discussion).

Also, somewhere in the methods needs to be added how the deer were aged, as this is included in the results.

Line 156:  When discussing the values in the shot deer, the higher value in some of the control group could suggest that the control group was not 100% fit and healthy and/or had suffered some stress, dehydration or other reason for elevated lactate.

Line 180: The other types of sedative if used (other than ACP) need to be discussed or at least mentioned.

Line 186: Comment - in the absence of evident gross injuries (as in category 2 animals), a high lactate might be treatable with aggressive fluid therapy and not require euthanasia in all cases. Should lactate levels, based on this paper, not perhaps be used in the decision-making process alongside consideration of injuries (clinical examination) and  availability of facilities (including a vet) for IV fluids and other supportive treatment?  Just would not want people to think that ALL deer in ALL circumstances  should be euthanised just because of a lactate level >10.2mmol/l

References: there is one missing! All need to be checked with the text numbering.

Reviewer 2 Report

Dear Authors,

Thank you for your work on this manuscript... I believe it may become a valuable contribution to the literature.

However, I have some significant concerns regarding your results and their interpretation, especially the statistical analysis:

First, I cannot tell from your manuscript if any significant differences were found in your work.  You reported that differences among groups were closed to significance?  I have not seen results reported like that before.  Does this mean that no significant differences were discovered?  If so, that means that no differences could be found between your groups.  Therefore you cannot conclude that Groups 1 and 2 are different than the other groups.  Yet, you state in your discussion that groups 1 and 2 appear similar supporting euthanasia in group 2.  You can speculate that there is a trend toward significance, but without the p-value that cannot be appreciated by the reader.

Also, it is important to include the p-values in your comparisons for the reader to be able to interpret your work.  In addition, it would be nice to include a bar graph or other method to compare the groups.  In this way, authors usually denote significant differences with different lower case letters next to each group. If this were done and no significance was found, then all groups would have the letter 'a' next to the group; if significance was found different letters (e.g. 'a' and 'b') would be used to denote significance among groups. 

Second, it looks like the ROC curves have allowed you to draw conclusions regarding the likelihood of roe deer survival following injury and admission.  I am not sure this is valid, as I believe the ROC analysis summarizes the performance of a classifier (in this case admission L-lactate), independent of the chosen cut-off. If the classifier (admission L-lactate) is not significant, then the data is insufficient to conclude that the classifier is useful at all (no matter what cut-off). 

Further, you have stated the cutoff value of 10.2 mmol/L, where animals with an admission L-lactate above this level would most likely die.  How does this cutoff value compare with other studies in wildlife or domestic animals?  As an example, this value compares well with elevated admission L-lactate values for horses with intestinal strangulation (e.g. Johnston K1, Holcombe SJ, Hauptman JG. Vet Surg. 2007 Aug;36(6):563-7. Plasma lactate as a predictor of colonic viability and survival after 360 degrees volvulus of the ascending colon in horses).

Again, a graph showing the ROC curves would help the reader understand how you were able to determine the L-lactate cutoff value of 10.2 mmol/L that helps determine survival from non-survival in admitted roe deer at a rescue center. 

Lastly, I have made some other comments and concerns in the manuscript (attached file with tracking changes) that should be addressed to help improve this manuscript.

Thank you again for your important contribution and work.

Reviewer 3 Report

The authors report the blood lactate concentration used as prognostic parameter to assess the outcome in roe deers. The paper is very interesting and in my opinion it could be suitable for pubblication after major revisions.

In general the paper needs an editing by an english native speaker becasue in this way it is twisted, in particular the discussion section and it could make not easier to understand the concepts in the discussion. Moreover, the authors need to improve the discussion section because it is not clear in many parts (too twisted sentences and not very good english).

Materials and methods. You assessed data distribution, then you used the krusla-wallis, thus I think the data did not show a gaussian distribution.. If yes, the results must be reported as median and standard error, minimum and maximum values. The median and standard deviation are used for gaussian distributed data. Please correct the table and the results and report only the results espressed as your statistical analysis allow you.

In the results you report an interval reference you found, but in the statistical analysis you did not report how you calculate this, you told just you have excluded the outliers, only in the results, but not in the amt and met section. Please add the statistical analysis for the calculation of the reference interval and the assessment of the outliers and then you can reportthem in the results section.

line 102: not kids; but fawns

lines 101-102: you reported the age in the result section, but you did nor report how you assess this parameter in the mat and met section. Please add this information in order to understand how you collect the data and then you can report in the results section.

Line 106: please change "espressed" with "reported".

Line 106-107: I know that 0.051 is really a "bad result" but the statistical analysis is significant or not and you are not allowed to report that "even if it is not significant it is more or less significant". If you want to have a significant results you need to add more animals and see what happens, but if the data you reported are these, the results is that you did not find statistical differences in blood lactate concentrations among group and this is the only results that you are allowed to report in a scientific report. So, please erase line 107 and wirte as follow: ... table 1. No statistical differences were found in blood lactate concentration among the groups (p=0.051).

For the same reason please erase line 111 in the talbe description.

Table: please change the comma with dots (english style) and please use the same decimal expression throughout the text (sometimes you report .00 and sometimes .0, please report all the numerical results in the same manner).

Moreover, please report the "name of the group" instead of the number in the table and erase the definition of the groups from the table caption. In this way tha table will be clearer. Moreover, on the side of the group name, please report the number of the animals you enrolled in each group. In this way it will be easier to understand the meaning of the last column on the right. In this column, please report the results as 8/xx (%).

all text: please change the comma with dots (english style) and please use the same decimal expression throughout the text (sometimes you report .00 and sometimes .0, please report all the numerical results in the same manner).

lines 113 and 117, please add mmol/L to the number reported in the tesx.

In my opinion the sensitivity you have chosen is too low, thus you will have false negative. This is your choice, thus if you want to keep on with this result, you must stressed in the evaluation of study limitations (better then you have already done).

lines 139-140: this sentence is not clear, please rephrase.

lines 146 and 147: makes uniform the decimals.

line 150: please erase that.

lines 146-152: in general this sentence is very confusing and it not clear at all the use of the reference intervals (more than one reported). Please try to rephrase the sentence to make it better interpretable.

lines 160-161: not clear, please rephrase.

line 162: appear, not appears.

lines 162-163: you cannot say this. As I reported earlier, the  statistical analysis is not an opinion! Please erase the sentence. The only thing you could say, and I think is the only with an important meaning, is that the dead and the humanely euthanized did not showed differences in blood lactate concentrations, thus it might be justify the euthanasia (only might because you support your hypothesis only with lactatemia, but you do not report others parameters to support this idea; thus a very lightweighted support).
